# A pilot randomised controlled trial comparing the effectiveness of the MaTerre180' participatory tool including a serious game versus an intervention including carbon footprint awareness-raising on behaviours among academia members in France

Claudia Teran-Escobar[1,2]*, Nicolas Becu[3]*, Nicolas Champollion[4], Nicolas Gratiot[1], Benoît Hingray[4], Gérémy Panthou[4], Isabelle Ruin[4]*

1 IGE, Univ. Grenoble Alpes, IRD, CNRS, INRAE, Grenoble INP, Grenoble, France, 2 Department of Psychology, University Paris Nanterre, Nanterre, France, 3 UMR 7266 LIENSs, CNRS, La Rochelle, France, 4 IGE, Univ. Grenoble Alpes, CNRS, IRD, G-INP, Grenoble, France

* cteranes@parisnanterre.fr (CTE); nicolas.becu@cnrs.fr (NB); isabelle.ruin@univ-grenoble-alpes.fr (IR)

**Data Availability Statement:** No datasets were generated or analysed during the current study. All

## Abstract

### Background

Activities embedded in academic culture (international conferences, field missions) are an important source of greenhouse gas emissions. For this reason, collective efforts are still needed to lower the carbon footprint of Academia. Serious games are often used to promote ecological transition. Nevertheless, most evaluations of their effects focus on changes in knowledge and not on behaviour. The main objectives of this study are to 1) Evaluate the feasibility of a control and an experimental behaviour change intervention and, 2) Evaluate the fidelity (the extent to which the implementation of the study corresponds to the original design) of both interventions.

### Methods

People employed by a French research organisation (N = 30) will be randomised to one of the two arms. The experimental arm consists in a 1-hour group discussion for raising awareness about climate change, carrying out a carbon footprint assessment and participating to a serious game called "Ma terre en 180 minutes." The control arm consists of the same intervention (1h discussion + carbon footprint assessment) but without participating to the serious game. On two occasions over one month, participants will be asked to fill in online surveys about their behaviours, psychological constructs related to behaviour change, sociodemographic and institutional information. For every session of intervention, the facilitators will assess task completion, perceived complexity of the tasks and the perceived responsiveness of participants. Descriptive statistics will be done to analyse percentages and averages of the different outcomes.

relevant data from this study will be made available upon study completion on https://osf.io/chpfk/?view_only=0ba1ecd434ce427fb115a447cf36c2c1 with the DOI https://doi.org/10.17605/OSF.IO/CHPFK.

**Funding:** The only funding received was from IRD (Institut de Recherche pour le Developpement) that funded the salary of the post-doctoral researcher (Cluadia Teran-Escobar). The funders had no role in study design, data collection and analysis, decision to publish, or preparation of the manuscript.

**Competing interests:** The authors declare no potential competing interests.

## Discussion

Ma-terre EVAL pilot study is a 1-month and a half pilot randomised controlled trial aiming to evaluate the feasibility and the fidelity of a 24-month randomised controlled trial. This study will provide more information on the levers and obstacles to reducing the carbon footprint among Academia members, so that they can be targeted through behaviour change interventions or institutional policies.

## Background

Reducing greenhouse gas emissions to successfully mitigate climate change is a great challenge of the 21st century. From demand-side perspective, an important percentage of greenhouse gases (72%) are related to household or "lifestyle" consumption such as housing, mobility and daily diet [1]. For this reason, citizens and communities are encouraged to change their behaviours and practices to mitigate climate change not only in their individual spheres but also in their professional and collective spheres [2]. If humanity wants to limit global warming to 1.5° (or 2°), a set of coordinated actions between citizens, investors, consumers, role models and professionals is needed [3].

Academia members, which include scholars, technical and administrative staff working in universities and research centres, are not exempt from contributing to greenhouse gas emissions and inequalities through certain behaviours. One common practice is attending in-person scientific conferences and field missions, which have been identified as significant sources of emissions [4]. Related to the Covid-19 lockdowns, many recent studies focused on conference habits in the academic sector. For example [5], compared the impact of three in-person conferences in the United States, Korea and England, and estimated that each academic attending to one of these three conferences produced between 1.3 and 1.8 tons of CO2e.

Moreover, research indicates that certain groups within academia face additional disadvantages in accessing in-person conferences. Women, individuals with household responsibilities, those requiring visa applications, scientists working in the Global South and early-career researchers are among those disproportionately affected [6–10].

Criticism has been raised both within and outside academia regarding the high emissions associated with such behaviours [11, 12], which can undermine the credibility of academics' recommendations on climate change [13]. Nonetheless, attending in-person scientific conferences continues to be strongly encouraged and often linked to academic excellence [7, 8, 14–16].

### Strategies for changing Academia's practices and behaviours: *Hard* and *soft* levers

Increasingly, universities and research centres (National Centre for Scientific Research [17], EHT Zurich [18]), communities of actors from the academia (Ma Terre en 180 minutes community [19], Labo1point5 community [20]) and international conference organisers [21] are committing to proposing strategies to reduce the carbon footprint of the academic world.

In this study protocol, strategies will be categorised in *Hard* levers and *Soft* levers [22–24]. *Hard* levers aim to change the economic and geographical contexts of individuals to encourage or discourage them from adopting a practice. In general, *hard* levers seek to build enabling environments for the adoption of behaviours [25]. In recent years, in the Academic context, *hard* levers as economic (dis)incentives (e.g., carbon tax on air travel [26]), limiting air travel

by applying individual or collective quotas [27, 28], replacing air travel by videoconference [7], prohibiting air travel (e.g., for domestic air travel or for easily achievable travel by train) and making it more complicate (e.g., longer procedures) to purchase airline tickets [29] have only started to be implemented.

*Soft* levers or behavioural interventions aim to change the psychological factors associated with behaviours, such as the intention to reduce air travel for professional mobility and subjective norms (the feeling of social pressure to adopt a practice deemed acceptable or expected by others). In general, *soft* levers seek to influence attitudes, motivations and reasons for adopting a practice [30]. In the field of behaviour change literature, these levers are referred to as behaviour change techniques [31]. Some examples of behaviour change techniques in the field of academic behaviours are the creation of tools to inform about the emissions of air travel [29], relaying detailed documents about how to travel in a train or engaging in eco-friendly actions during conferences (e.g., [21]), informing about the social and environmental impacts of air travelling [9], developing and promoting the use of collaborative workshops and serious games to raise awareness and discuss transition pathways [19]. Indeed, serious games are becoming increasingly widespread to promote pro-environmental behaviours and climate change adaptation behaviours [for reviews, read 32–34]. Although the use of serious games can influence the perception of climate change issues, stimulate cognitive engagement and even increase the intention of adopting new behaviours [32, 33], most of them present similar limitations [32]. First, most studies examine and evaluate the gains in knowledge and participants' engagement without measuring any actual behavioural changes. Second, they often lack of rigorous methodologies including controlled designs (i.e., comparing the effects of an experimental group containing a serious game versus a control group containing other content or no content at all) and longitudinal designs (i.e., studies following the participants' behaviours and knowledge over weeks or months). Third, to our knowledge, there is a lack of studies taking into account the socio-spatial context of the individuals who participate in serious games (their age, their socio-economic status, the characteristics of their living areas). Ma Terre-EVAL study aims to address all the aforementioned limitations.

## Ma Terre-EVAL pilot study

Ma Terre-EVAL is a study carried out by a consortium of researchers in social psychology, human geography and climate science aiming to better understand the motivations and obstacles to adopt ecological transitions paths in Academia. For this purpose, Ma Terre-EVAL combines concepts and knowledge in climate science, gaming and simulation, human geography and behavioural sciences to evaluate the effects of a serious game on three academic practices: professional mobility by air, daily commuting by car and digital and material purchases. Moreover, the motivations towards changing behaviours and the socio-economic and institutional contexts will also be assessed. Ma Terre-EVAL will rely on rigorous methodologies such as a randomised controlled study comparing an experimental arm versus a control arm, a longitudinal follow-up of 6 weeks spread over 24 months and a large-scale study with 1000 participants. More precisely, the sample size was estimated by using *a priori* sample calculation in G*Power 3.1.9.4 [35, 36] by considering that implementing *soft* levers in our second dependent variable (because no study has investigated the effects of *soft* levers on air travel) reduces 7% of daily travel by car ($g$ = 0.16, Z = 6.419, p < .001, 95% CI [0.113, 0.213], [37] and considering a significance level of 0.05 and a statistical power of 80%. The behaviours of Academia's actors will be assessed through detailed online surveys (e.g., questionnaires about the numbers of academic travels, distances travelled and questionnaires assessing intentions, habits, subjective norms and socio-economic and institutional contexts).

The main objective of this pilot study will be to evaluate the feasibility and the fidelity of Ma Terre-EVAL study through a shorter version (a follow-up of two weeks over one month and a half) with a small number of participants. In this study, the feasibility will be understood as the extent to which the different elements of the intervention and evaluation are feasible for the implementation team [38] and the fidelity will be understood as the extent to which all the elements of the intervention are accurately implemented and without modifying the behaviour change techniques and contents [39]. For evaluating the feasibility and the fidelity of this study, a 1-month and a half pilot randomised controlled trial, parallel groups, two arms, exploratory trial with a 1:1 allocation ratio will be implemented. More precisely, the objectives of the study are twofold:

a.  Evaluate the feasibility of the different elements of the interventions (completion or non-completion of planned tasks, assessment of task complexity, participation rates in intervention sessions) and the evaluation (completion rate of online questionnaires and quality of the responses) proposed in Ma Terre-EVAL study.

b.  Evaluate the fidelity of the behavioural interventions provided to the experimental and control arms (the extent to which all the different elements of the interventions and evaluation are feasible for the team implementing the study).

## Methods

The Table 1 shows the SPIRIT schedule of the present study. All the details related to the tools used for the assessments are presented in Table 2.

### Ethics and data protection

The Fig 1 shows the procedure of ethics, data protection, recruitment, inclusion in the study, allocation, study enrolment and beginning of the study. First, the main researchers of this study contacted the Data Protection officer to establish a detailed planned of data collection and data protection. Then, a detailed document containing the data collection, data analysis and data protection were submitted to the Grenoble Alpes Research Ethics Committee (CERGA). Ma Terre-EVAL pilot study received the ethic's approval in May 2023 (File CERGA-Avis-2023-11). Moreover, each participant will read and sign a detailed informed consent form (cf. S1 and S2 Appendices.) before starting the study only if he/she agrees to participate in the study.

### Participants

**Study setting.**   This study will be restricted to individuals being employed by a French private or public research organisation. In France, 462k (i.e., considering that, in total 649 1000 people work partial or full time for R&D representing 462K full-time works) of people work full time for a French private or public research organisation [45]. According to a recent survey involving more than 6000 people working in public research centres [20], between 25% and 35% of the respondents have travelled by air for professional reasons between 2017 and 2019. The principal motives of professional air mobility were conferences, meetings, workshops and research visits. According to this same survey, 24% of the respondents commute by car to go to work.

**Eligibility criteria.**   To take part in Ma Terre-EVAL pilot study, participants will need to meet the following criteria:

**Table 1. The recommendation of interventional trials (SPIRIT) schedule of enrolment, interventions, and assessments.**

| | STUDY PERIOD | | | | | |
| --- | --- | --- | --- | --- | --- | --- |
| | Enrolment | Allocation | Post-allocation | | | Close-out |
| TIMEPOINT | $-S_1$ | 0 | $S_0$ | $SI_1$ | $SI_2$ | $S_1$ |
| **ENROLMENT:** | | | | | | |
| **Eligibility screen** | X | | | | | |
| **Informed consent** | X | | | | | |
| **Allocation** | | X | | | | |
| **INTERVENTIONS:** | | | | | | |
| *Intervention A/Experimental Arm* | | | | X | X | |
| *Intervention B/Control Arm* | | | | X | | |
| **ASSESSMENTS:** | | | | | | |
| *Feasibility of the intervention and adherence* Task completion, Perceived complexity of tasks, Participation in sessions | | | | X | X | |
| *Feasibility of the evaluation* Completion rate of online surveys, Quality of responses | | | X | | | X |
| *Fidelity of the intervention* Exposure to the intervention, Perceived responsiveness of participants | | | | X | X | |
| *Academic's behaviours* Number of professional trips by air, Percentage of home-work trips made by car or other modes, Frequency of purchasing of equipment, Use of alternatives to air travel | | | | X | | |
| *Psychological constructs* Intention towards changing behaviours, Knowledge about climate change Beliefs about the consequences of behaviour on the climate, Self-efficacy, Beliefs about negative consequences of reducing air travel, Attitude, Environmental attitudes, Ecological identity, habits, Compensatory beliefs, Self-control resources, Descriptive and subjective norms | | | X | | | X |
| *Sociodemographic and institutional information* Gender, age, number of children, self-reported distance between home and work, perceived accessibility by train or alternative transport modes, income level, Career status, employer, characteristics, academic recognition, funding for professional mobility | | | | | | |

Note. S = Session (Survey to be fill up), SI = Session if intervention.

- Being over 18 years old

- Being employed by a French public or private research organisation.

## Participant timeline

**Recruitment, allocation, and blinding.** The participants will be recruited through announcements by e-mail sent by the study partners (CNRS, IRD, Univ. Grenoble-Alpes) and "snowball". The study partners (CNRS, IRD, Univ. Grenoble-Alpes) will relay announcements describing the study and the steps to follow to participate.

The people volunteering to participate in Ma Terre-EVAL pilot study will send an email to the field manager who will call them back to explain the study procedure and verify the eligibility criteria.

Participants accepting to participate will be randomly allocated to the experimental or control arm. For this purpose, previously to the enrolment, the scientific team will create a randomisation list by blocks of ten (i.e., in this randomisation list by blocks of ten, here are five possibilities to be in the experimental arm and five possibilities to be in the control arm).

Concerning the blinding, the implementation team (the facilitators in charge of leading the intervention sessions) will not be aware of the arm allocation at the beginning of the study ($S_0$), but blinding will be impossible afterward because the facilitators deliver different contents to each arm through online meetings with the participants. The research team might be aware of the group allocation at the moment of analysing the data of the pilot study because of the

**Table 2. Summary of the variables, tools and time measurements.**

| Outcomes | Tool | $S_0$ | $SI_1$ | $SI_2$ | $S_1$ |
|---|---|---|---|---|---|
| **Main outcomes of the study** | | | | | |
| **Feasibility of the intervention and adherence** | | | | | |
| Task completion | Cheklist of tasks to complete | | ✔[a] | ✔[a] | |
| Perceived complexity of tasks | Scale of 1 to 3 (easy to difficult) | | ✔[a] | ✔[a] | |
| Participation in sessions | Rate of participation/absence in sessions | | ✔[a] | ✔[a] | |
| **Feasibility of the evaluation** | | | | | |
| Completion rate of online surveys | Percentage of completion of online questionnaires | ✔[b] | | | ✔[b] |
| Quality of responses | Duration of each questionnaire (too short or too long times considered low quality) | ✔[b] | | | ✔[b] |
| **Fidelity of the intervention** | | | | | |
| Exposure to the intervention | Duration of each session | | ✔[a] | ✔[a] | |
| Perceived responsiveness of participants | Scale of 1 to 7 (not at all responsive to very responsive) | | ✔[a] | ✔[a] | |
| **Secondary outcomes** | | | | | |
| **Academic's behaviours** | | | | | |
| Number of professional trips by air | Survey | ✔ | | | |
| Percentage of home-work trips made by car et by active and sustainable mobility | Survey | ✔ | | | |
| Frequency of purchasing of equipment | Survey | ✔ | | | |
| Intention towards changing professional trips made by car, reducing home-work trips made by car and frequency of purchasing of equipement | Adapted survey [40] | ✔ | | | ✔ |
| Use of alternatives to air travel | Survey | ✔ | | | |
| **Psychological constructs** | | | | | |
| Knowledge about climate change | Multiple choice questionnaire | ✔ | | | ✔ |
| Beliefs about the consequences of behaviour on the climate | Adapted survey [9] | ✔ | | | ✔ |
| Self-efficacy of choosing an alternative to air travel | Adapted survey [40] | ✔ | | | ✔ |
| Self-efficacy of choosing an alternative to car travel | Adapted survey [40] | ✔ | | | ✔ |
| Self-efficacy of making sustainable purchases | Adapted survey [40] | ✔ | | | ✔ |
| Beliefs about negative consequences of reducing air travel | Adapted survey [20] | ✔ | | | ✔ |
| Attitudes towards alternatives to air travel | Adapted survey [40] | ✔ | | | ✔ |
| Attitudes towards alternatives to car travel | Adapted survey [40] | ✔ | | | ✔ |
| Environmental attitudes | Environmental attitudes scale [41] | ✔ | | | ✔ |
| Ecological identity | Green identity scale [42] | ✔ | | | |
| Business travel habits by air | Adapted habits scale [43] | ✔ | | | |
| Commuting habits by car | Adapted habits scale [43] | ✔ | | | |
| Compensatory beliefs | Survey [9] | ✔ | | | ✔ |
| Self-control resources | Subjective vitality scale [44] | ✔ | | | ✔ |
| Descriptive and subjective norms | Adapted survey [40] | ✔ | | | ✔ |
| **Sociodemographic and institutional information** | | | | | |
| Socio-demographic questionnaire (Gender, age, number of children, self-reported distance between home and work, perceived accessibility by train or alternative transport modes, income level) | Survey | ✔ | | | |
| Socio-demographic follow-up questionnaire (number of children, self-reported distance between home and work, perceived accessibility by train or alternative transport modes, income level) | Survey | | | | ✔ |
| Employment questionnaire (Career status, employer characteristics, academic recognition, funding for professional mobility) | Survey | ✔ | | | |

*(Continued)*

**Table 2.** (*Continued*)

| Outcomes | Tool | $S_0$ | $SI_1$ | $SI_2$ | $S_1$ |
|---|---|---|---|---|---|
| Employment follow-up questionnaire (career status, employer characteristics, academic recognition, funding for professional mobility) | Survey | | | | ✔ |

**Note.** By default, the outcomes will be measured by using an online survey. S = Session (Survey to be fill up), SI = Session if intervention, R = Recruitment and enrolment periods.

[a] indicates that the outcome was measured by a questionnaire by paper

[b] indicates that the outcome was calculated by a member of the research team based on other data.

missing data (because the control arm will have no information about the second intervention meeting). Nevertheless, the lack of blinding will not be a problem, because the objective of this study is not to determine the effects of the intervention but the feasibility and the fidelity of the intervention and the evaluation.

**Timeline of the study.** Participants enrolled in this study will have a first 45-minutes online meeting with a member of the implementation team ("Session 0" or $S_0$, see Fig 2). At the beginning of this meeting, the participant will read and electronically sign and send by mail the consent form (cf. S1 and S2 Appendices). Then, he or she will fill in a questionnaire to report on his or her professional mobility, home-to-work mobility and professional electronic purchases, psychological constructs and socio-economic and institutional information (see Table 2 for all the details about the tools and surveys and see S3 and S4 Appendices).

Approximately two weeks after this session, participants of both experimental and control arm will have the first intervention meeting ($IM_1$) with another four participants of their same arm and a facilitator (1 hour). During this meeting, the participants will have a discussion aiming to raise awareness between academic behaviours and environmental issues (all the details of this session will be described in the section "Intervention"). At the end of this meeting, participants of both arms will be asked to evaluate their carbon footprint through the Micmac app (https://avenirclimatique.org/micmac/simulationCarbone.php) and send it to the implementation team.

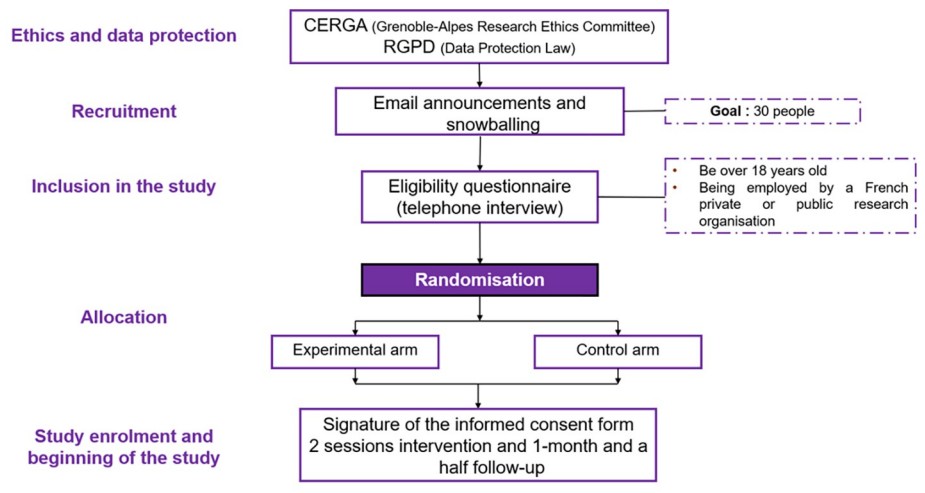

**Fig 1. Procedure of the Ma Terre-EVAL pilot study since the ethics and data protection procedures to the study enrolment and beginning of the study.**

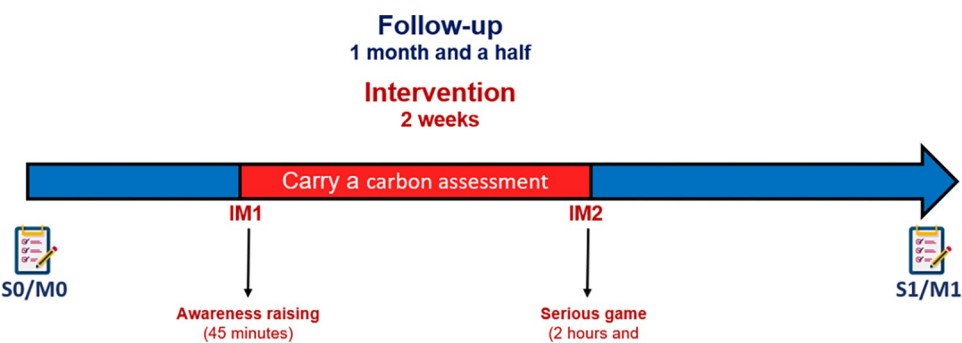

**Fig 2. Calendar of Ma Terre-EVAL pilot study and the measurements.** S = Session (Survey to be fill up),
M = Month, IM1 = Intervention meeting with experimental and control arm. IM2 = Intervention meeting only with
experimental arm. Notebook = Online survey.

Approximately two weeks after this first intervention meeting, only the participants of the experimental arm will be contacted to carry out a second intervention meeting ($IM_2$) with four participants of their same arm and a facilitator (2 hours and 20 minutes). During this meeting, the participants will participate to the serious game "Ma Terre en 180 minutes" aiming to discuss the possible transition pathways to diminish carbon emissions while playing the role of fictional research laboratory team members (all the details of this session will be described in the section "Experimental Arm"). During and after each of the meetings ($IM_1$ and $IM_2$) the facilitators will complete a checklist and answer questions about the sessions (see Table 2 and S1–S3 Tables. Tables to be filled up by the facilitators to evaluate de fidelity and feasibility).

Two weeks after the experimental arm's intervention meeting session (IM2) and one month after the control arm's intervention meeting session (IM2), the participants will receive a link to complete a 15-minutes online survey ("Session 1" or "$S_1$").

**Retention of participants.** To limit the attrition risk, we will send a newsletter one month after the start of the study to inform about the laboratories involved in the study and a short presentation of the research and field team.

**Power analysis and sample size.** Because our data analysis will be descriptive as a part of the feasibility study, we consider that a power analysis is not necessary [38]. Considering time and resources constraints, we decided to recruit thirty participants ($N = 30$) that will be randomised in the experimental or control arm.

## Intervention

Eligible participants will be randomised in equal proportion to the experimental arm or to the control arm. The intervention for the experimental arm will last for two weeks (including two intervention meetings and a carbon assessment) as the one for the control arm (including only one intervention meeting and a carbon assessment of the one month and a half study period) (Fig 2). The detailed content of the interventions for experimental and control arms will be described in the next subsections (see Table 3 for a summary).

**Experimental arm.** The intervention (Intervention A) to be carried out by this arm consists of an awareness-raising session (one hour in a group of five people with a facilitator), the completion of a carbon footprint assessment and a "Ma Terre en 180 minutes" game session (2 hours 20 minutes in a group of five people with a facilitator).

During the first online 1-hour meeting (Intervention meeting 1, $IM_1$), the facilitator and the group of five participants will discuss about the energy consumption and the basic human's

**Table 3. Summary of the elements of the experimental and the control arms.**

| Element | Experimental arm | Control arm |
|---|---|---|
| First group appointment with a facilitator | Group discussion about topics related to planetary limits, climate change, distribution of the carbon footprint of some research laboratories and initiatives to decrease the Academia's carbon footprint. | Group discussion about topics related to planetary limits, climate change, distribution of the carbon footprint of some research laboratories and initiatives to decrease the Academia's carbon footprint. |
| Conducting an individual carbon assessment | Online individual carbon assessment related to food, mobility and energy consumption. | Online individual carbon assessment related to food, mobility and energy consumption. |
| Second group appointment with a facilitator | Group session of the serious game "Ma terre en 180 minutes". During the serious game, each participant takes on the role of two fictional characters from a research laboratory whose mission is to reduce the laboratory's carbon footprint by 50%. | |

needs, planetary limits, climate issues, current carbon footprint of a random French person and target carbon footprint according to the Paris Agreement, the link between activities in an academic life and carbon footprint, the distribution of the carbon footprint of some laboratories and the initiatives to diminish it. At the end of the meeting, they will collectively make an astonishment report using virtual Post-its. The behavioural change techniques [31] that were targeted by these activities are "5.2 Salience of consequences", "5.3 Information about social and environmental consequences", "6.3 Information about others' approval" and "4.1 Instruction on how to perform the behaviour".

After the first meeting of the intervention, the study participants are encouraged to carry out an online carbon footprint assessment by using a free tool: https://avenirclimatique.org/micmac/simulationCarbone.php. This takes between ten and fifteen minutes and requires precise information about the household energy consumption. The behavioural change technique [31] that was targeted by this activity is "2.2 Self-monitoring of behaviour".

During the second online 2 hour and 20 minutes meeting (Intervention meeting 2, $IM_2$), the group of five participants play the serious game "Ma terre en 180 minutes" (see Fig 3 to see

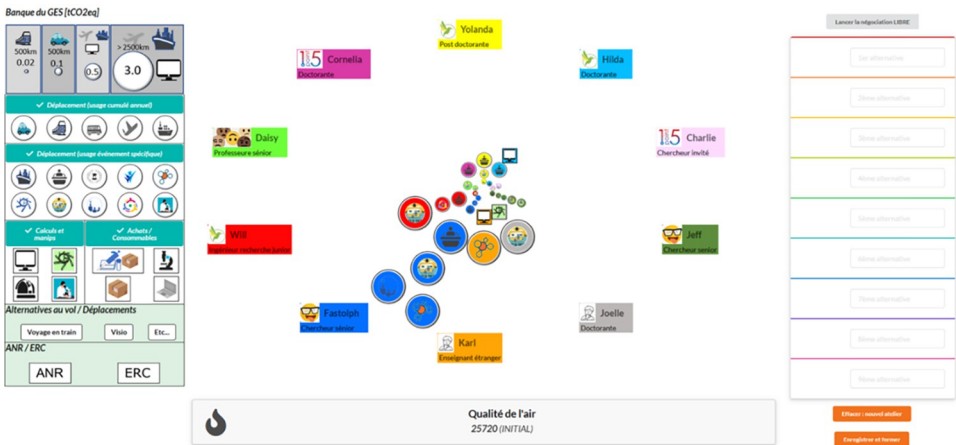

**Fig 3. Example of a game board from the serious game "Ma terre en 180 minutes".** The tokens show the different sources of carbon footprint of each fictional character (e.g., travel for conferences or field missions, commuting). The bigger tokens indicate bigger carbon footprints.

an example of a game table or http://51.178.55.78/MT180/mt180.htm) under the gaze of the facilitator who answers questions and paces the game steps. During the game session, each player takes on the role of two characters from a research team and begins a free negotiation followed by a guided negotiation aiming at a 50% reduction of the research laboratory carbon footprint. In addition, after the end of the game, the facilitator leads a debriefing session with the participants. The behavioural change techniques [31] that was targeted by the serious game and its debriefing are "6.2 Social comparison", "5.2 Salience des consequences", "13.2 Framing/reframing", "6.1 Demonstration of the behaviour", "4.1 Instruction about how to perform a behavior", "12.2 Restructuring the social environment". Moreover, the detailed content of the intervention is available on S5 and S6 Appendices.

**Control arm.**   The intervention (Intervention B) to be carried out by this arm consists of an awareness-raising session (one hour in a group of five people with a facilitator) and the completion of a carbon footprint assessment.

During the first online 1-hour meeting (Intervention meeting 1, $IM_1$), the facilitator and the group of five participants will discuss about the energy consumption and the basic human's needs, planetary limits, climate issues, current carbon footprint of a random French person and target carbon footprint according to the Paris Agreement, the link between activities in an academic life and carbon footprint, the distribution of the carbon footprint of some laboratories and the initiatives to diminish it. At the end of the meeting, they will collectively make an astonishment report using virtual Post-its. The behavioural change techniques [31] that were targeted by these activities are "5.2 Salience of consequences", "5.3 Information about social and environmental consequences", "6.3 Information about others' approval" and "4.1 Instruction on how to perform the behaviour".

After the first meeting of the intervention, the study participants are encouraged to carry out an online carbon footprint assessment by using a free tool: https://avenirclimatique.org/micmac/simulationCarbone.php. This takes between ten and fifteen minutes and requires precise information about the household energy consumption. The behavioural change technique [31] that was targeted by this activity is "2.2 Self-monitoring of behaviour". The control group does not participate in the serious game "Ma Terre en 180 minutes".

The Figs 4 and 5 compares the targeted sources of behaviour (i.e., motivation, opportunity or capability according to [31]), the intervention functions (i.e., education, persuasion, incentivisation, coercion, training, enablement, modelling, environmental restructuring and restriction according to [31]) and the behaviour change techniques [31] mobilised in the Experimental and Control arms.

**Adherence.**   To measure the adherence to each intervention arm, the facilitator will take notes about the duration of each intervention meeting and of every situation that could disrupt the planned elements of the intervention (e.g., participants not sending their carbon footprints, participants lacking an intervention meeting).

## Outcomes and data collection

**Primary outcomes and data collection methods.**   The main outcomes of Ma Terre-EVAL pilot will be the feasibility of the intervention, the feasibility of the evaluation of behavioural changes and the fidelity of the intervention.

The intervention's feasibility will be operationalised in three variables: 1) Task completion (a checklist of all activities that need to be carried out during the intervention sessions), 2) Complexity of the intervention tasks (a scale of 1 to 3, from easy to difficult) and 3) Percentage of participation in the meetings (percentage of people absent or present at the intervention meetings). All of this information will be collected through paper questionnaire.

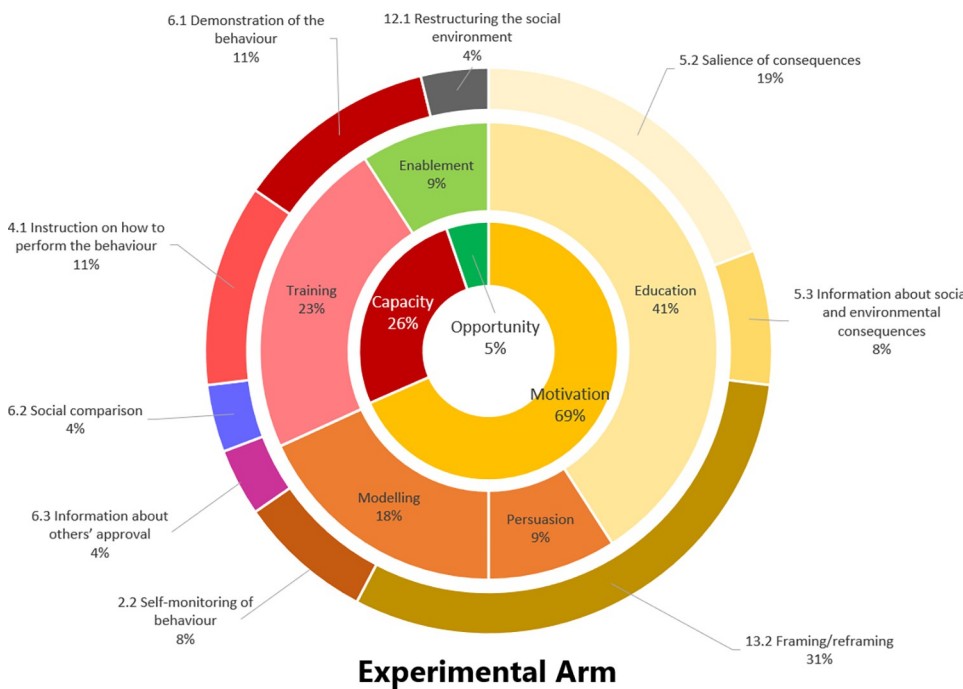

**Fig 4. Sources of behaviour targeted by the intervention (in the centre of the circle), intervention functions (in the intermediary part) and behaviour change techniques (in the exterior part) used for the experimental arm.**

The feasibility of the evaluation of behavioural change will be operationalised in two variables: 1) Completion rate of the online questionnaires (average of the completion of each questionnaire) and 2) Quality of the answers (considering the duration of each questionnaire).

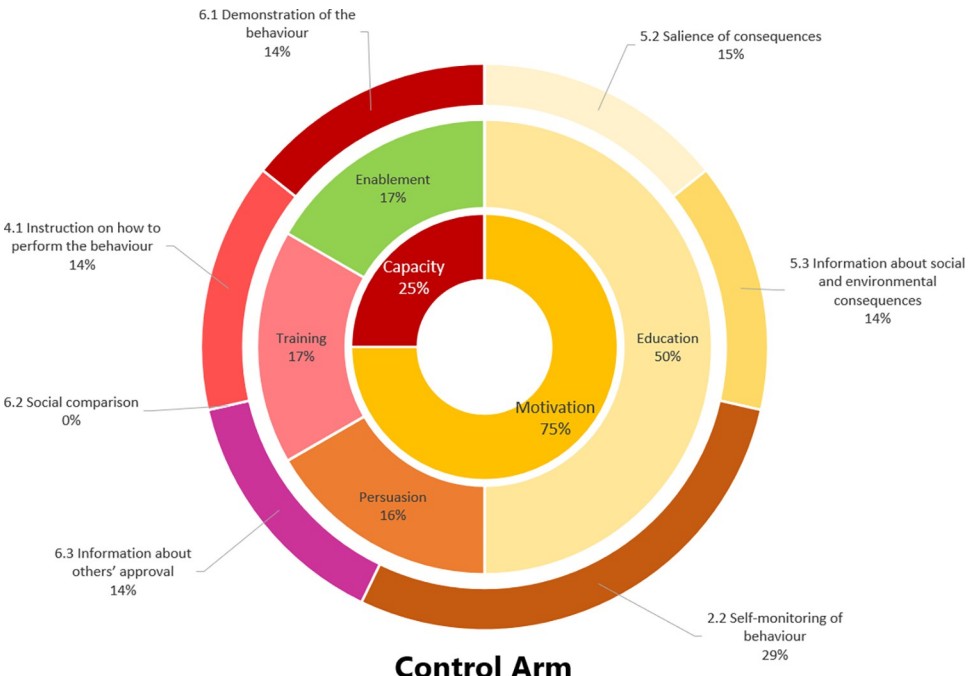

**Fig 5. Sources of behaviour targeted by the interventions (in the centre of the circle), intervention functions (in the intermediary part) and behaviour change techniques (in the exterior part) used for the control arm.**

More precisely, the quality of the answers will allow us to detect participants who potentially do not take the task seriously enough (i.e., with unrealistic short time on certain tasks) by calculating the Median Absolute Deviation [46] to detect outliers on the time spent on the page showing the article and will exclude participants with a Median Absolute Deviation superior to 3. Both of these outcomes will be calculated by the scientific team by downloading the online surveys completed by the participants of the study.

Finally, the fidelity of the intervention will be operationalised in two variables: 1) Exposure to the intervention (average time of each session) and, 2) Responsiveness of the participants (via a scale of 1 to 7 from not at all responsive to very responsive, this question will be asked to each facilitator). All of this information will be collected through paper forms. All the details of the variables, tools and measurement times can be found in Table 2.

**Secondary outcomes and data collection methods.** Secondary outcomes of Ma Terre-EVAL pilot will include the following variables related to Academia's behaviours: Number of professional trips by air, percentage of home-to-work trips made by car, by active and sustainable mobility, frequency of numerical equipment purchase, intention towards changing professional trips made by car, reducing home-to-work trips made by car and frequency of numerical equipment purchase equipment [40] and the use of alternatives to air travel through online surveys implemented on the platform Sphinx iQ2 v 7.3.1.0. Moreover, we will collect psychological constructs such as knowledge about climate change (measured through a multiple-choice questionnaire), beliefs about the consequences of behaviour on the climate [9], self-efficacy of choosing an alternative to air travel [40], self-efficacy of choosing an alternative to car travel [40], self-efficacy of making sustainable purchases [40], beliefs about negative consequences of reducing air travel [20], attitudes towards alternatives to air travel [40], attitudes towards alternatives to car travel [40], environmental attitudes [41], ecological identity [42], professional travel habits by air [43], commuting habits by car [43], compensatory beliefs [9], perceived self-control resources [44], descriptive and subjective norms [40] through online surveys. Finally, socio-demographic and institutional information such as gender, age, number of children, self-reported distance between home and work, perceived accessibility by train or alternative transport modes, income levels, career status, employer characteristics, perceived academic recognition and, funding for professional mobility. All the details of the variables, tools and measurement times can be found in Table 2.

## Data management and statistical methods

**Data quality, management, storage, access and confidentiality.** In order to monitor the quality of the data, one member of the scientific team will check the documents filled in by the facilitators (i.e., the documents evaluating the feasibility and the fidelity) and the presence of missing data in these documents after the end of each intervention meeting. This data will be immediately transferred to an Excel table. Furthermore, data from online surveys will be downloaded at the end of each session of measurement.

The data will be stored in two blocks:

Block 1 will contain the contact file including name and email of the participants for sending the surveys and setting the intervention meetings. Moreover, this block will contain the correspondence table between the participants' identifier and their name. This block will be encrypted and it will only be accessible by the scientific coordinator and the field coordinator.

Block 2 will contain all the collected data from the paper forms and the online surveys. The data will be anonymised (only containing the participant's identifier).

Both of the blocks will be locally stored on the secured data centre of the university Grenoble-Alpes with access restricted only to Ma Terre-EVAL pilot team.

**Data monitoring, harms and auditing.** No data monitoring committee has been conformed for this study because the researchers have no strong suspicion that any of the interventions can potentially harm the participants. The coordinator of the implementation team and the coordinator of the scientific team will meet once every two weeks to audit the trial conduct.

**Ancillary and post-trial care.** No ancillary and post-trial care will be provided.

**Trial registration.** Because of the non-clinical character of this research study, we have submitted a registration in https://www.protocols.io/ with the registration number YXMVM24JBG3P.

**Statistical methods.** *Analysis of the feasibility of the intervention and adherence.* Concerning the Task completion, we will identify the tasks that were not completed (i.e., for each group of participants and for each meeting) and we will calculate the percentage of completion of each task (e.g., the discussion about planetary limits was completed in 100% of $IM_1$ for experimental and control arms). Concerning the perceived complexity of tasks, we will calculate the average complexity of each task (from 1 to 3) and look at all the comments noted by the facilitators. Concerning the adherence, a participation/absence rate will be done per each intervention meeting.

*Analysis of the feasibility of evaluation.* Concerning the Completion rate of online surveys, we will calculate the percentage of completion of online questionnaires during the first session of measurement ($S_0$) and the second session of measurement ($S_1$). Concerning the quality of the responses, we will calculate the Median Absolute Deviation [46] to detect outliers that spent too few or too much time to complete the surveys (participants with a Median Absolute Deviation $> 3$).

*Analysis of the fidelity of the intervention.* Concerning the exposure to the intervention, we will calculate the average duration of each intervention meeting. Concerning the perceived responsiveness of participants, we will calculate the average of perceived responsiveness per each intervention meeting.

## Discussion

Ma-terre EVAL pilot study is a 1-month and a half pilot randomised controlled trial designed to evaluate the feasibility and the fidelity of a 24-month randomised controlled trial aiming to change academic behaviours such as professional mobility by air, daily commuting and digital and material purchases. Ma-terre EVAL pilot study will include a two-arm intervention of 2-weeks: an experimental arm that includes an awareness-raising group discussion, an online carbon footprint assessment and a group session of the serious game "Ma Terre en 180 minutes" [19]; and a control arm which includes only the awareness-raising group discussion and the online carbon footprint assessment. We believe that because the control arm targets mainly raising awareness about climate change and professional behaviours (through the group discussion and the carbon footprint assessment) and consequently it will target sources of behaviour such as capacity and motivation (by using behavioural techniques such as 2.2 self-monitoring of behaviour, 5.3 information about social and environmental consequences, 5.2 salience of consequences according to [31]). Furthermore, the experimental arm will additionally aim the co-construction of ecological transition paths in Academia (through the serious game) and potentially will lead to capacity and opportunity building (by using additional behavioural techniques such as 13.2 framing/reframing, 12.1 restructuring the social environment and 6.2 social comparison according to [31]).

Moreover, the evaluation of the effects of Ma Terre-EVAL pilot study includes two sessions of online surveys to measure the participants' behaviours (number of professional trips by air, percentage of home-to-work trips made by car and by active and sustainable mobility, frequency of purchasing of equipment), the intention towards changing professional trips made

by car, reducing home-to-work trips made by car and frequency of purchasing of equipment, psychological mechanisms related to changing behaviours (attitudes, subjective norms, self-efficacy, ecological identity, habits) and socio-demographic and institutional information (e.g., gender, age, income levels, career status, perceived academic recognition).

Because of the complexity of the interventions proposed to the experimental and control arms and the complexity of the evaluation of the behavioural changes, this pilot randomised controlled trial aims to evaluate the feasibility (the extent to which all the different elements of the interventions and evaluation are feasible for the team implementing the study) of the intervention and the evaluation of the fidelity (the extent to which all the elements of the intervention are accurately implemented and without modifying the behaviour change techniques and contents). Indeed, several authors [39, 47, 48] have indicated that every large-scale study should be pre-tested with small groups of participants to identify elements that could be adapted to ensure better feasibility and fidelity.

## Supporting information

**S1 Checklist.**
(DOC)

**S1 Appendix. Information notice and informed consent—experimental group.**
(DOCX)

**S2 Appendix. Information notice and informed consent—control group.**
(DOCX)

**S3 Appendix. Online survey–sessions 0, 2, 3, 4 and 5.**
(DOCX)

**S4 Appendix. Online survey–session 1.**
(DOCX)

**S5 Appendix. Content of the intervention provided to the control group.**
(DOCX)

**S6 Appendix. Content of the intervention provided to the experimental group.**
(DOCX)

**S1 Table. Checklist for the feasibility of the first session of intervention.**
(DOCX)

**S2 Table. Checklist for the feasibility of the second session of intervention.**
(DOCX)

**S3 Table. Checklist for participation to the sessions.**
(DOCX)

## Acknowledgments

The authors thank the IRD (Institut for Research and Developpement).

## Protocol version and protocol amendments

This version of the protocol (1.0) is dated 1 May 2023. Any changes in the protocol will be communicated and shared on https://www.protocols.io/view/my-earth-eval-pilot-yxmvm 24jbg3p/v1 with the DOI: 10.17504/protocols.io.yxmvm24jbg3p/v1.

## Author Contributions

**Conceptualization:** Claudia Teran-Escobar, Nicolas Becu, Nicolas Champollion, Nicolas Gratiot, Benoît Hingray, Gérémy Panthou, Isabelle Ruin.

**Investigation:** Isabelle Ruin.

**Methodology:** Claudia Teran-Escobar, Isabelle Ruin.

**Project administration:** Claudia Teran-Escobar, Nicolas Becu, Nicolas Gratiot.

**Supervision:** Nicolas Becu, Isabelle Ruin.

**Validation:** Nicolas Gratiot.

**Writing – original draft:** Claudia Teran-Escobar.

**Writing – review & editing:** Nicolas Becu, Nicolas Champollion, Nicolas Gratiot, Benoît Hingray, Gérémy Panthou, Isabelle Ruin.

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
