## [Decision Letter · Decision Letter 0]

18 Sep 2023

PONE-D-23-23933A pilot randomised controlled trial comparing the effectiveness of the MaTerre180’ participatory tool including a serious game versus an intervention including carbon footprint awareness-raising on behaviours among academia members in FrancePLOS ONE

Dear Dr. Teran-Escobar,

Thank you for submitting your manuscript to PLOS ONE. After careful consideration, we feel that it has merit but does not fully meet PLOS ONE’s publication criteria as it currently stands. Therefore, we invite you to submit a revised version of the manuscript that addresses the points raised during the review process.

Adjust the paper according previewers comments.==============================

A marked-up copy of your manuscript that highlights changes made to the original version. You should upload this as a separate file labeled 'Revised Manuscript with Track Changes'.An unmarked version of your revised paper without tracked changes. You should upload this as a separate file labeled 'Manuscript'.If applicable, we recommend that you deposit your laboratory protocols in protocols.io to enhance the reproducibility of your results. Protocols.io assigns your protocol its own identifier (DOI) so that it can be cited independently in the future. For instructions see: https://journals.plos.org/plosone/s/submission-guidelines#loc-laboratory-protocols. Additionally, PLOS ONE offers an option for publishing peer-reviewed Lab Protocol articles, which describe protocols hosted on protocols.io. Read more information on sharing protocols at https://plos.org/protocols?utm_medium=editorial-email&utm_source=authorletters&utm_campaign=protocols.

We look forward to receiving your revised manuscript.

Kind regards,

Radoslaw Wolniak, full professor

Academic Editor

PLOS ONE

3. We note that Figure 4 in your submission contain copyrighted images. All PLOS content is published under the Creative Commons Attribution License (CC BY 4.0), which means that the manuscript, images, and Supporting Information files will be freely available online, and any third party is permitted to access, download, copy, distribute, and use these materials in any way, even commercially, with proper attribution. For more information, see our copyright guidelines: http://journals.plos.org/plosone/s/licenses-and-copyright.

A. You may seek permission from the original copyright holder of Figure 4 to publish the content specifically under the CC BY 4.0 license. 

B. If you are unable to obtain permission from the original copyright holder to publish these figures under the CC BY 4.0 license or if the copyright holder’s requirements are incompatible with the CC BY 4.0 license, please either i) remove the figure or ii) supply a replacement figure that complies with the CC BY 4.0 license. Please check copyright information on all replacement figures and update the figure caption with source information. If applicable, please specify in the figure caption text when a figure is similar but not identical to the original image and is therefore for illustrative purposes only.

Reviewers' comments:

Reviewer's Responses to Questions

**Comments to the Author**

1. Does the manuscript provide a valid rationale for the proposed study, with clearly identified and justified research questions?

Reviewer #1: Yes

Reviewer #2: Yes

2. Is the protocol technically sound and planned in a manner that will lead to a meaningful outcome and allow testing the stated hypotheses?

Reviewer #1: Partly

Reviewer #2: Yes

3. Is the methodology feasible and described in sufficient detail to allow the work to be replicable?

Reviewer #1: Yes

Reviewer #2: Yes

4. Have the authors described where all data underlying the findings will be made available when the study is complete?

Reviewer #1: Yes

Reviewer #2: Yes

5. Is the manuscript presented in an intelligible fashion and written in standard English?

Reviewer #1: Yes

Reviewer #2: Yes

6. Review Comments to the Author

You may also provide optional suggestions and comments to authors that they might find helpful in planning their study.

Reviewer #1: Dear Authors

I accepted the paper. In my opinion yours paper is good. The topic is acctual and the problem is presnted clear.

Reviewer #2: Overall, the article is correct. Paper is also current in scientific and utilitarian terms. Analysis and conclusions are correct. Accurate and pertinent literature. The abstract clearly outlines the main objectives and presents the methods. The background is well-written. The authors effectively convey the need to limit global warming by influencing individual behaviors, as well as those in professional and collective spheres. They particularly focus on academic communities and their carbon footprint. In my opinion, this is a very interesting topic. The authors identify and explain the research gaps.

Potential improvements:

- Please consider shortening the title to a maximum of 9 words. A more concise version of the proposed title would be preferable.

- I suggest adding a comment under Table 1 on page 9. Although there is no Figure 1, the authors mention it on page 8 (Methods section). The methods section is well-written. The authors fully explain the methodology procedure. Authors precisely present details on figures which is strength of this paper.

- In the summary, clarify who may find the results useful and for what purpose. Describe the significance of the research and its impact on a broader field, demonstrate how the acquired information can be further utilized.

7. PLOS authors have the option to publish the peer review history of their article (what does this mean?). If published, this will include your full peer review and any attached files.

Reviewer #1: No

Reviewer #2: No

---

## [Author Response · Author response to Decision Letter 0]

9 Jan 2024

Dear Reviewers, 

Thank you very much for the time you devoted to our article and for your encouraging and positive comments. . Your comments have helped us to improve and refine the arguments and added value of our study.

Reviewer 2 :

Comment: Please consider shortening the title to a maximum of 9 words. A more concise version of the proposed title would be preferable.

Answer: Unfortunately, the SPIRIT guidelines for clinical trials (Chan et al., 2013) indicate that the Title should containt the study design, population, interventions, and, if applicable, trial acronym. For this reason, the title is long.Nevertheless, I changed the short-title to have fewer words.

Comment: In the summary, clarify who may find the results useful and for what purpose. Describe the significance of the research and its impact on a broader field, demonstrate how the acquired information can be further utilized.

Answer: I modified the summary to include this information in page 3, lines 44-46.

Comment: I suggest adding a comment under Table 1 on page 9. Although there is no Figure 1, the authors mention it on page 8 (Methods section). 

Answer: We agree that it was not really clear that Figure 1 is actually a table describing SPIRIT guidelines information, we modified the title of the figure (page 9, line 162)

---

## [Decision Letter · Decision Letter 1]

12 Mar 2024

A pilot randomised controlled trial comparing the effectiveness of the MaTerre180’ participatory tool including a serious game versus an intervention including carbon footprint awareness-raising on behaviours among academia members in France

PONE-D-23-23933R1

Dear Dr. Teran-Escobar,

We’re pleased to inform you that your manuscript has been judged scientifically suitable for publication and will be formally accepted for publication once it meets all outstanding technical requirements.

Kind regards,

Zhihua Zhang

Academic Editor

PLOS ONE

Additional Editor Comments (optional):

Reviewers' comments:

Reviewer's Responses to Questions

**Comments to the Author**

1. Does the manuscript provide a valid rationale for the proposed study, with clearly identified and justified research questions?

Reviewer #1: Yes

2. Is the protocol technically sound and planned in a manner that will lead to a meaningful outcome and allow testing the stated hypotheses?

Reviewer #1: Yes

3. Is the methodology feasible and described in sufficient detail to allow the work to be replicable?

Reviewer #1: Yes

4. Have the authors described where all data underlying the findings will be made available when the study is complete?

Reviewer #1: Yes

5. Is the manuscript presented in an intelligible fashion and written in standard English?

Reviewer #1: Yes

6. Review Comments to the Author

You may also provide optional suggestions and comments to authors that they might find helpful in planning their study.

Reviewer #1: PLOS ONE,

Review, 25 Feb. 2024,

PONE-D-23-23933R1

Dear Authors,

Your manuscripts was accepted by me. After all reviewers and your improvements , it should be published.

I have only one note: Fig 5. I propose Fig. 5 a and Fig 5 B, The sentence (page 22) “A pilot to evaluate the effect of a participatory tool on Academic’s low-carbon behaviours” should be in text, no between figs. In the text , you should write after the sentence (Fig. 5 b) (if I see good), and in the previous sentence (Fig.5a).

Best wishes

Reviewer.

7. PLOS authors have the option to publish the peer review history of their article (what does this mean?). If published, this will include your full peer review and any attached files.

Reviewer #1: No

---

## [Editor Report · Acceptance letter]

19 Mar 2024

PONE-D-23-23933R1 

PLOS ONE

Dear Dr. Teran-Escobar, 

I'm pleased to inform you that your manuscript has been deemed suitable for publication in PLOS ONE. Congratulations! Your manuscript is now being handed over to our production team.

Kind regards, 

on behalf of

Dr. Zhihua Zhang 

Academic Editor

PLOS ONE